# Comparison of Three Algorithms for Predicting Infarct Volume in Patients with Acute Ischemic Stroke by CT Perfusion Software: Bayesian, CSVD, and OSVD

**DOI:** 10.3390/diagnostics13101810

**Published:** 2023-05-20

**Authors:** Yunzhuo Yao, Sirun Gu, Jiayang Liu, Jing Li, Jiajing Wu, Tianyou Luo, Yongmei Li, Bing Ge, Jingjie Wang

**Affiliations:** 1Department of Radiology, The First Affiliated Hospital of Chongqing Medical University, No. 1 Youyi Road, Yuzhong District, Chongqing 400016, China; 2Medical Imaging Center, Central Hospital of Shaoyang, Shaoyang 422000, China; 3NO. 958th Hospital of PLA Army, Chongqing 400020, China; 4Canon Medical Systems Clinical Scientific Department, No. 162 North District Road, Yuzhong District, Chongqing 400016, China

**Keywords:** bayesian, CT perfusion, infarct core, ischemic stroke, singular value decomposition

## Abstract

This study aimed to compare the performance of the Bayesian probabilistic method, circular Singular Value Decomposition (cSVD), and oscillation index Singular Value Decomposition (oSVD) algorithms in Olea Sphere for predicting infarct volume in patients with acute ischemic stroke (AIS). Eighty-seven patients suffering from AIS with large vessel occlusion were divided into improvement and progression groups. The improvement group included patients with successful recanalization (TICI 2b-3) after thrombectomy or whose clinical symptoms improved after thrombolysis. The progression group consisted of patients whose clinical symptoms did not improve or even got worse. The infarct core volume from the Olea Sphere software was used as the predicted infarct volume (PIV) in the improvement group, whereas the hypoperfusion volume was used as the PIV in the progression group. We defined predicted difference (PD) as PIV minus final infarct volume (FIV) measured at follow-up imaging. Differences among the three algorithms were assessed by the Friedman test. Spearman correlation analysis was used to verify the correlation between PIV and FIV. In addition, we performed a subgroup analysis of the progression group based on collateral circulation status. The median [interquartile range (IQR)] of the PD and Spearman correlation coefficients (SCCs) between PIV and FIV for the improvement group (n = 22) were: Bayesian = [6.99 (−14.72, 18.99), 0.500]; oSVD = [−12.74 (−41.06, −3.46), 0.423]; cSVD = [−15.38 (−38.92, −4.68), 0.586]. For the progression group (n = 65), the median (IQR) of PD and SCCs were: Bayesian = [1.00 (−34.07, 49.37), 0.748]; oSVD = [−0.17 (−53.42, 29.73), 0.712]; cSVD = [66.55 (7.94, 106.32), 0.674]. The Bayesian algorithm in the Olea Sphere software predicted infarct volumes with better accuracy and stability than the other two algorithms in both the progression and improvement groups.

## 1. Introduction

As reported in previous studies, rapid reperfusion significantly reduces mortality and functional dependency for patients with acute ischemic stroke (AIS) [1,2,3]. Timely reperfusion therapy is the top choice for AIS patients whose onset time is within six hours [4]. However, for patients beyond the time window, whether to choose reperfusion therapy is primarily a weighing of risk and benefit based on the size of the infarct core and penumbra [5,6]. Patients would take a greater risk for complications from reperfusion therapy due to excessive infarct core volume [7]. Therefore, it is important to accurately assess the core infarct volume in these patients.

Multiple clinical trials, as represented by DAWN and DEFUSE 3 trials, manifested that incorporating perfusion imaging findings into screening criteria can improve neurological outcomes in patients with AIS after endovascular treatment [4,8,9]. Studies have demonstrated that diffusion-weighted imaging (DWI) provides the most accurate assessment of early infarction in most cases, but is rarely used in clinical practice due to the potential time delay [10,11]. CTP has a diagnostic value close to that of MRI in identifying ischemic core and can achieve rapid imaging [12,13], and thus it can help select patients suitable for endovascular treatment.

Currently, various CTP post-processing algorithms are used to calculate the infarct core and penumbra [14,15,16], and the advanced algorithm provides quantitative data over the physician’s visual assessment. However, this also presents a problem for clinical decision-making when multiple calculations are inconsistent. Olea Sphere (Olea Medical, La Ciotat, France) is a CTP post-processing application that is widely used in various hospitals and research institutions [17,18,19], providing three algorithms: the Bayesian probabilistic method, circular Singular Value Decomposition (cSVD), and oscillation index Singular Value Decomposition (oSVD). This study aimed to compare performances of the Bayesian, oSVD, and cSVD algorithms at the recommended threshold within Olea in predicting the infarct volume of AIS patients, using final infarct volume (FIV) as the reference value. Considering that collateral circulation is an independent predictor of outcomes in patients with AIS [20], we performed a subgroup analysis of the progression group based on collateral circulation in further.

## 2. Materials and Methods

### 2.1. Patients

As a monocentric, retrospective, and observational study, we consecutively involved 158 AIS patients in our hospital between June 2020 and November 2021. A total of 87 patients satisfying the following criteria were included: (1) age > 18 years; (2) AIS with unilateral internal carotid artery (ICA), M1 or M2 middle cerebral artery (MCA), A1 or A2 anterior cerebral artery (ACA) severe stenosis or occlusion; (3) CTA/CTP scan on admission and follow-up NCCT or DWI after 24 h; (4) complete demographic, clinical, and laboratory data. Exclusion criteria: (1) posterior circulation stroke; (2) inapplicable NCCT or MRI follow-up images; (3) substandard CT and MRI scans. We were granted a waiver of informed consent by the institutional ethics committee. The screening process for enrolled patients is shown in Figure 1.

Eighty-seven patients were divided into the progression group (n = 65) and the improvement group (n = 22). The improvement group included those who received intravenous thrombolysis and whose condition improved, or who underwent thrombectomy or thrombolysis–thrombectomy bridging therapy and then underwent successful reperfusion (m TICI 2b-3). Patients in the progression group either did not receive any of the intervention treatments above, or, even if they received intervention, experienced no improvement or worsening of their condition (based on the discharge NIHSS score or comprehensive evaluation before discharge by attending clinicians).

In addition, considering the independent predictive value of collateral circulation for condition progression, patients in the progression group were divided into two subgroups according to the ASITN/SIR collateral grading scale modified based on dynamic CTA and time-MIP technology to score secondary collateral (Poor—0/1/2; Abundant—3/4).

### 2.2. Imaging Technology and Analysis

All the patients included underwent a one-stop CTA/CTP scan on a 320-row detector CT scanner (Aquilion ONE, Canon Medical Systems Corporation, Otawara, Japan) on admission. In the process of scanning, dynamic volume one-shot scan mode was first performed to acquire NCCT images, then the whole brain dynamic volume intermittent mode was used to perform CTA and CTP examination. Dynamic volume perfusion scanning is automatically initiated at 7 s after contrast agent injection to ensure simultaneous contrast perfusion and scanning. NCCT, CTA, CTP, and related data were obtained by 2-s intermittent scanning in the arterial phase and 5-s intermittent scanning in the venous phase, and the total collection time was 60 s. Scan parameters of the CT scanner were as follows: 80 kV, 150–310 mA, coverage of 140–160 mm, 1.0 mm slice thickness, and 1.0 mm interval to improve reconstructive speed, reconstruction with adaptive iterative dose reduction. The total dose of scanning was 5.0–6.0 mSv (k = 0.0021). The quantity and velocity of the contrast medium were automatically calculated according to the patient’s sex, height, weight, and concentration of contrast medium, and the contrast medium was injected using the P3T high-pressure syringe technique. (Iopamidol, Bracco Sine, Milan, Italy). Figure 2A–D show the admission NCCT and 4D-CTA images of one patient in progression group.

### 2.3. CTP Analysis

The original CTP data were imported into Olea Sphere 3.0-SP28, and the volume of the infarct core and hypoperfusion area was automatically delineated and calculated by three threshold algorithms. The perfusion parameters to be used included relative cerebral blood flow [rCBF], time to peak [TTP], absolute Tmax [aTmax], and difference Tmax [dTmax]. This software automatically recognizes and processes perfusion images and uses the local AIF method to obtain an average value of arterial input function (AIF) and venous output function (VOF). The thresholds of the three post-processing algorithms we used were as follows: (1) default Bayesian estimation algorithm, rCBF < 25% and dTTP > 5 s for infarct core together with dTTP > 5 s for hypoperfusion area; (2) cSVD algorithm, rCBF < 40% and aTmax > 2 s for infarct core together with aTmax > 6 s for hypoperfusion area; (3) oSVD algorithm, rCBF < 45% and dTmax > 4 s for infarct core together with dTmax > 4 s for hypoperfusion area. The CTP post-processing images of one patient in progression group are shown in Figure 2E–G.

### 2.4. Follow-Up NCCT or DWI MRI Analysis

NCCT or DWI MRI images at follow-up were collected and used to manually delineate the FIV through ITK-SNAP software. The infarct area was defined as the low-density region of the affected cerebral hemisphere on each layer of the NCCT images or the high-intensity region on the DWI MRI images. DWI MRI is preferred for FIV. FIV delineation was consensually performed by two neuroradiologists together. Figure 2H shows the follow-up DWI image of one patient in progression group.

### 2.5. Statistical Analysis

According to the previous study, we hypothesized all penumbra would convert to infarct in patients in the progression group, whereas all penumbra would be salvaged in patients in the improvement group [21]. Patients in the progression group were included to assess the volume of all hypoperfusion areas, whereas patients in the improvement group were included to assess the volumes of infarct core volume. Therefore, infarct core volume predicted by three algorithms was chosen for the improvement group as the predicted infarct volume (PIV) that would be compared with FIV, whereas the predicted hypoperfusion volume was chosen for the progression group. Patients in the progression group were further divided into two subgroups: Subgroup 1.a (n = 29) included patients with conservative treatment and abundant collaterals using predicted infarct core volume as PIV; Subgroup 1.b (n = 29) included patients with conservative treatment and abundant collaterals using predicted hypoperfusion volume as PIV; Subgroup 2 (n = 36) included patients with conservative treatment and poor collateral condition using predicted hypoperfusion volume as PIV. We defined predicted difference (PD) as equal to PIV minus FIV. The positive difference indicates the overestimation of FIV, whereas the negative value indicates the underestimation. We conducted the difference analysis and correlation analysis between FIV and both predicted infarct core volume and hypoperfusion area volume in patients with abundant collaterals, and similar analyses were performed between FIV and predicted hypoperfusion area in patients with poor collateral condition.

Measurement data was described as mean ± SD or median (IQR). Categorical data were described as rate. As our measurement data did not conform to the normal distribution, the Mann–Whitney U test was used for the comparison of measurement variables, and the Chi-square test was used for counting data. The Friedman test was used to compare the difference between FIV and infarct value measured by the Bayesian algorithm, oSVD algorithm, and cSVD algorithm in each group, to determine which algorithm was more reliable. In each group, the proportion of patients with overestimated FIV was used to indirectly reflect the stability of the algorithm. Scatter plots and Spearman correlation analysis were conducted to show the correlation between FIV and volume predicted by three algorithms for each group. SPSS version 22.0 (IBM, Armonk, NY, USA) was used for the above statistical analysis. *p*-values < 0.05 were considered to be statistically different.

## 3. Results

The demographic and clinical data predicted infarct volume and final infarct volume of all patients are shown in Table 1. MCA was the majority of responsible vessels in both the improvement group and the progression group before and after grouping (Figure 3A–C). Large artery atherosclerosis (LAA) was the majority of Trial of Org 10172 in Acute Stroke Treatment (TOAST) types before grouping or in the progression group after grouping, whereas cardioembolism (CE) was the majority in the improvement group after grouping (Figure 3D–F).

There was a statistical difference in time from onset to admission (<6 h and >24 h), hypertension, atrial fibrillation, dyslipidemia, and follow-up infarct volume between groups (*p* < 0.05).

### Infarct Volumetric Analysis

The comparison of PD was calculated from the three algorithms of AIS patients in both the improvement and the progression group (Figure 4A). There were statistical differences between the three algorithms regardless of group (*p* < 0.001 and *p* = 0.002). For the improvement group, the PD of the Bayesian algorithm was smaller [Median (IQR), 6.99 (−14.72, 18.99)], compared with oSVD (*p* = 0.014) and cSVD (*p* = 0.003). In addition, there was no significant difference between oSVD [Median (IQR), −12.74 (−41.06, −3.46)] and cSVD [Median (IQR), −15.38 (−38.92, −4.68); *p* = 1.000]. As for the progression group, the predicted difference of Bayesian [Median (IQR), 1.01 (−34.07, 49.37)] algorithm was smaller, compared with cSVD [Median (IQR); 2.21 (−32.66, 17.31); *p* < 0.001] and oSVD [Median (IQR), −0.17 (−32.66,17.31); *p* < 0.001]. Furthermore, there was a statistically significant difference between oSVD and cSVD (*p* = 0.018) in the progression group. 

The comparison of PD calculated from different PIVs among the two subgroups is shown in Figure 4B. In the subgroup 1.a, the PD of the Bayesian algorithm was the smallest [Median (IQR), −11.22 (−51.31, −0.68)] and had a statistical difference from that of cSVD (*p* < 0.001). The PD of oSVD [Median (IQR), −16.24 (−77.96, −5.16)] had no statistical difference from that of cSVD [Median (IQR), −19.49 (−82.07, −4.95)] (*p* = 0.147) or the Bayesian algorithm (*p* = 0.054). In the subgroup 1.b, the PD of the Bayesian algorithm [Median (IQR); 2.96 (−14.13, 35.51)] and the oSVD algorithm [Median(IQR); 2.49 (−11.29, 28.80)] were both small, but there was no statistical difference between them (*p* = 1.000). Both of them had statistical differences from cSVD [Median (IQR); 43.12 (4.73, 98.18), *p* < 0.001]. The PD of conservative treatment with abundant collaterals using hypoperfusion area volume from the Bayesian algorithm was closer to FIV than that using infarct core volume from the Bayesian algorithm (*p* < 0.001). In subgroup 2, the PD of the Bayesian algorithm [Median (IQR); 0.88 (−45.47, 56.22)] was the smallest, which has statistical difference from oSVD [Median (IQR), −9.04 (−71.22, 30.59), *p* = 0.002] and cSVD [Median (IQR), 73.76 (13.08, 110.56), *p* < 0.001]. There was a statistical difference between the PD of oSVD and that of cSVD (*p* < 0.001).

The proportion of patients with overestimated infarct is exhibited in Table 2.

The correlation between the FIV and PIV of the three algorithms was described through Spearman’s correlation coefficients (SCCs) and is shown in scatter plots (Figure 5A–F). The correlation analysis for progression subgroups based on collateral circulation status is included in the Appendix A. The PIV of most groups showed a correlation with FIV (*p* < 0.05).

In the progression group, the PIV of all three algorithms correlated with FIV (*p* < 0.001) among which the SCC of the Bayesian algorithm was 0.748. In the improvement group, the PIV of all three algorithms also correlated with FIV (*p* < 0.05) among which the SCC of the Bayesian algorithm was 0.500. Meanwhile, a few groups were showing a weak correlation between PIV and FIV: the SCCs of oSVD in subgroup 1.a and the improvement group were 0.397 and 0.423. Nevertheless, the PIV and FIV showed no correlation of cSVD in subgroup 1.a (*p* = 0.783). 

## 4. Discussion

This study compared the performance of Bayesian, oSVD, and cSVD algorithms under the recommended thresholds of the Olea Sphere CTP software in predicting infarct volume in patients with AIS. We identified the more accurate of the three algorithms in predicting FIV through this study so that clinicians can choose this algorithm for future quantitative evaluation of infarct core and penumbra in emergency CTP examinations. In the following, we will discuss the main findings.

For the progression group, according to the box plots and Spearman correlation analysis, the PIV from the Bayesian algorithm had a smaller predicted difference and a higher correlation with FIV. This result suggests that the Bayesian algorithm is more accurate in predicting the final infarct volume for patients whose infarcts progressed. Regarding the results of the improvement group, the PIV of all three algorithms was moderately correlated with FIV. Among them, the Bayesian algorithm also had a smaller difference between PIV and FIV. This result suggests that the Bayesian algorithm is more accurate in predicting the final infarct volume for patients whose penumbra has been salvaged. Furthermore, according to the proportion of patients with overestimated infarct (Table 2), the Bayesian algorithm can predict FIV robustly within two groups. In contrast, the SVD algorithms tend to underestimate the core while overestimating the hypoperfusion region, especially the cSVD. This may lead to intervention treatment in patients who do not have appreciable penumbra in clinical practice, which would increase the incidence of EVT complications. Thus, the Bayesian algorithm has a significant advantage in calculating infarct core and penumbra volume and is a worthy choice for clinicians.

A recent study mentioned that the estimation of ischemic core volume by the Bayesian algorithm is more stable than estimation by the SVD algorithm, which is consistent with our findings and may be related to the different delayed sensitivity of the Bayesian algorithm and SVD algorithm to tracer arrival [22]. In AIS events due to large vessel occlusion, the hypoperfusion region could be compensated to some extent by recruiting collateral circulation, whereas the infarct core with feeble collaterals has nearly no inflow of blood. The TDC of the infarct core tends to present as a flat shape, similar to the appearance of noise. At this point, the effective signal-to-noise ratio within the voxel is reduced. Therefore, the calculation with SVD algorithms, which are sensitive to noise, will result in a relatively large bias. However, the Bayesian algorithm can weaken this bias as it is not sensitive to noise, which allows for a more stable estimation of hemodynamic parameters [23].

For the progression group, we performed a subgroup analysis according to the status of collateral circulation. In patients with favorable collateral circulation, the relationship with FIV was compared when using the ischemic core (Subgroup 1.a) and the hypoperfusion zone (Subgroup 1.b) as PIV, respectively. The PIV of Subgroup 1.b had smaller differences (Figure 4B) and better correlation (Appendix A) compared with Subgroup 1.a, implying that it is more appropriate to select the low perfusion region to predict the final infarct volume in patients with AIS who do not achieve recanalization, regardless of the collateral circulation. This may suggest that abundant collaterals did not significantly reverse the conversion of penumbra to infarction for patients who have not achieved recanalization. Some investigators supported evidence that penumbral salvage and infarct growth are less time-dependent and more contingent on collateral flow [24]. According to this view, the reference value for FIV should be the infarct core area on CTP. Other researchers reported that collaterals sustained activity of brain tissue only for a limited period so the ischemic penumbra was destined to degenerate into infarction [6]. According to this view, the hypoperfusion area volume on CTP should be selected as the predicted infarct volume. The results of our subgroup analysis supported the latter view. However, we did not control for the factor of onset time due to sample size limitations, which may lead to bias in the statistical analysis, and this point needs to be further verified with a larger sample of studies.

According to the results of this study, the Bayesian algorithm is capable of predicting the infarct core with relative accuracy, but there are still some unavoidable factors that lead to bias, as follows. Previous studies have suggested the presence of ghost infarct core, which means a reduced volume at follow-up DWI compared with admission DWI [25]. Similarly, it is reasonable that the infarct core predicted by emergency CTP is not exactly equal in volume to the follow-up FIV, even when the penumbra is completely salvaged. In addition, part of the infarct core was already in the subacute or chronic phase at the time of follow-up DWI, and the high signal was not evident, which may lead to underestimation of the infarct core volume in some cases. However, since the same patients were used in the evaluation of each algorithm, the possibility of any impact caused by the above inevitable factors was the same in each algorithm, and thus this kind of bias could be eliminated. 

Notably, the range of predicted difference in the progression group is larger than that in the improvement group, as Figure 4A suggests. This may be mainly because there are more factors affecting the prediction of the penumbra. It is known that there is still no independent method to evaluate the penumbra volume precisely. CTP post-processing software calculated volume only based on the threshold of the opposite hemisphere but cannot evaluate the individual patient’s specific hemodynamics and other clinical characteristics. Tissues with a naturally low vascular flow rate in one hemisphere can be displayed as penumbra even if they function normally [26,27]. Hence, the volume of the hypoperfusion area and penumbra may be overestimated. In addition, spontaneous recanalization may also bring about an overestimation of the hypoperfusion area volume in this study [28]. Moreover, patients in the progression group had significantly larger infarct cores and FIVs overall than those in the improvement group, and an equal degree of bias would show up as a larger difference in a sample with a larger base size. Thus, according to the boxplot, the performance of all three algorithms does not seem especially good in the progression group. 

Our study has some limitations. This study was a retrospective single-center analysis with a comparatively small sample size, especially in the improvement group, which may affect the accuracy and credibility of the results, and thus the subgroup analysis of the improvement group based on lateral branch status was not implemented. First, due to the retrospective nature of the study, the time from onset to admission imaging and the time from CTA/CTP to follow-up imaging were not strictly controlled; second, the follow-up volume may be impacted by subjective factors due to manual rather than machine mapping. During the estimation of Olea software, color patches located in obviously wrong locations (e.g., ventricles, extracerebral tissue) were manually erased, but no precise spatial alignment was performed. 

## 5. Conclusions

The Bayesian algorithm used with its appropriate threshold is more accurate and stable than SVD algorithms in assessing FIV regardless of whether the condition of patients will improve or not. The status of collateral circulation had little effect on the selection of the algorithm. 

## Figures and Tables

**Figure 1 diagnostics-13-01810-f001:**
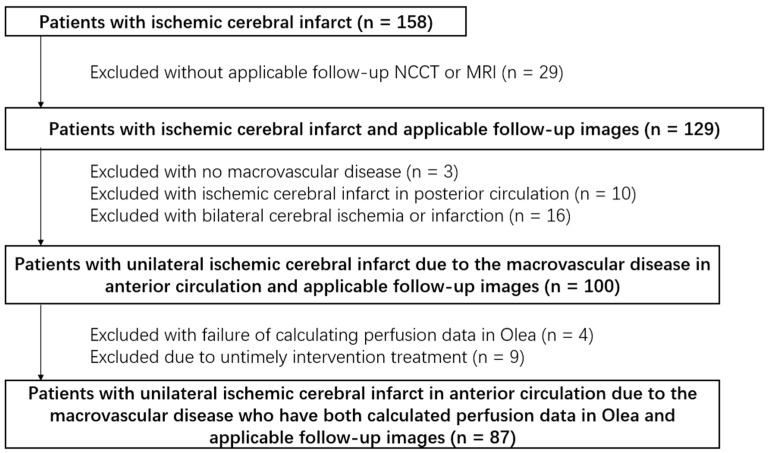
Screening flow chart of enrolled patients.

**Figure 2 diagnostics-13-01810-f002:**
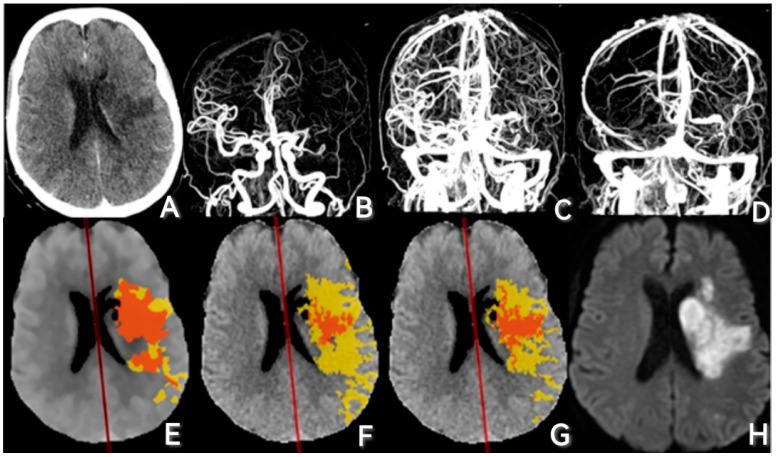
Example of one patient in progression group. (**A**) is the NCCT on admission of a 65-year-old female who has occlusion of the left middle cerebral artery (MCA). The FIV of this patient is 66.77 mL. (**B**–**D**) exhibit the early arterial phase, arterial phase, venous phase, and late venous phase in dynamic cerebral angiography of the patient mentioned above with 4D-CTA technology. After grading, this patient received a 2 for the collateral circulation status. (**E**–**G**) show the volume analysis of the same patient, which was calculated by the Bayesian algorithm (red:33.193 mL; yellow:67.517 mL), cSVD (red:8.042 mL; yellow:96.835 mL), and oSVD (red:14.182 mL; yellow:66.534 mL), respectively, in Olea Sphere post-processing software. Infarct core volume is shown in red and hypoperfusion area volume is shown in yellow. The overlapped area is shown in orange. Since the infarct core is always contained in the hypoperfusion area, only orange and yellow can be seen in the image. (**H**) is the follow-up DWI of the same patient.

**Figure 3 diagnostics-13-01810-f003:**
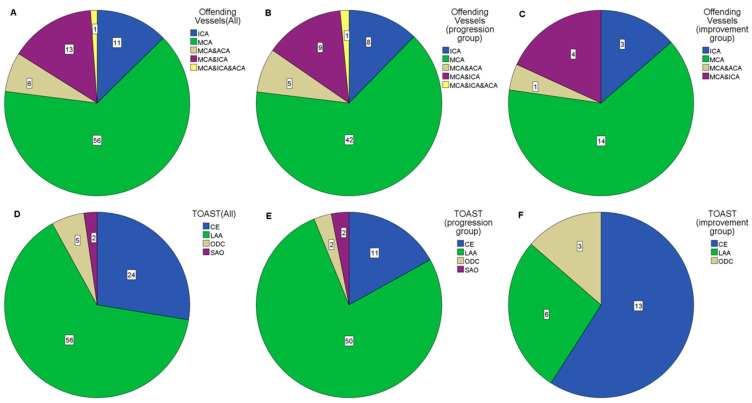
The proportion of offending vessels and TOAST in groups. The proportion of responsible vessels and the TOAST types in the enrolled patients are shown in different color blocks by pie charts as above, the number of cases of each type is put on the corresponding color block. (**A**,**D**) for all enrolled patients, (**B**,**E**) for the progression group, and (**C**,**F**) for the improvement group.

**Figure 4 diagnostics-13-01810-f004:**
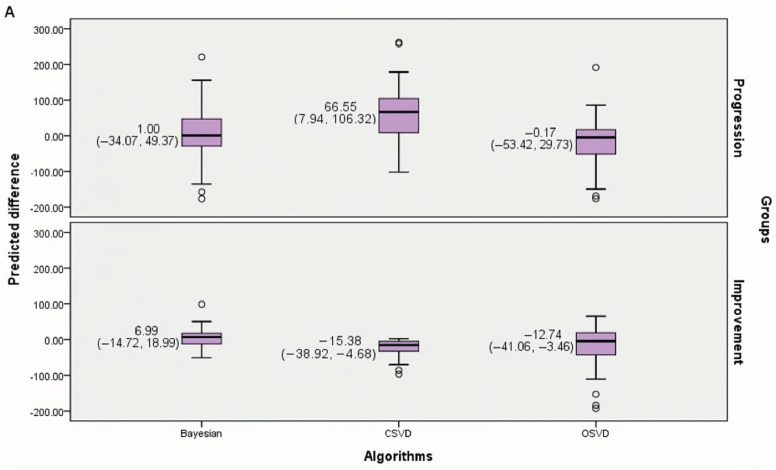
The Predicted Difference Calculated from the Three Algorithms. The difference between PIV predicted by the three algorithms and FIV is shown in the boxplot, and the corresponding medians and quartiles are attached. (**A**) is for the progression group and the improvement group. (**B**) is for the subgroup of the progression group.

**Figure 5 diagnostics-13-01810-f005:**
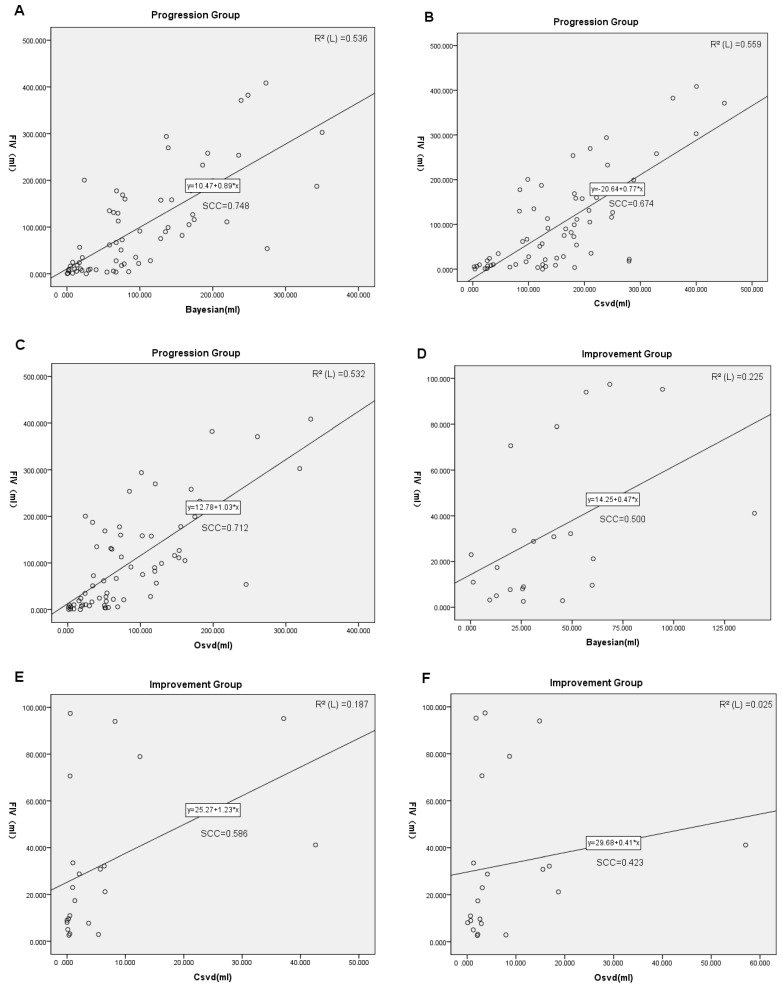
The correlation between PIV and FIV of the three algorithms. The results of correlation analysis are shown by scatter plots for each algorithm in different groups as above, and corresponding SCCs are shown on each plot; (**A**–**C**) for the progression group and (**D**–**F**) for the improvement group. FIV, final infarct volume; SCC, Spearman correlation coefficient; cSVD, cyclic singular value decomposition algorithms; oSVD, oscillatory exponential singular value decomposition algorithms.

**Table 1 diagnostics-13-01810-t001:** Demographic, clinical characteristics, and outcomes of patients with ischemic stroke.

Characteristic	All (n = 87)	Progression Group (n = 65)	Improvement Group (n = 22)
Male sex (%)	50 (50/87)	40 (40/65)	10 (10/22)
Age (years) (mean ± SD) (median) (IQR)	67.805 ± 13.542 (69.0) (59.0, 78.0)	67.446 ± 12.056 (68.0) (58.0, 77.0)	68.864 ± 17.507 (76) (67.0, 80.0)
NIHSS score (mean ± SD) (median) (IQR)	9.563 ± 7.378 (10) (2.0, 15.0)	8.815 ± 7.295 (9.0) (2.0, 14.5)	11.773 ± 7.342 (11) (7.25, 16.00)
Risk factors			
Smoke (%)	40 (40/87)	32 (32/65)	8 (8/22)
Hypertension (%)	64 (64/87)	52 (52/65)	12 (12/22)
Diabetes (%)	30 (30/87)	22 (22/65)	8 (8/22)
Coronary heart disease (%)	15 (15/87)	12 (12/65)	3 (3/22)
Atrial fibrillation (%)	26 (26/87)	14 (14/65)	12 (12/22)
Dyslipidemia (%)	41 (41/87)	36 (36/65)	5 (5/22)
Hemorrhagic transformation (%)	19 (19/87)	11 (11/65)	8 (8/22)
Time from onset to admission			
<6 h(%)	36 (36/87)	18 (18/65)	18 (18/22)
6–24 h(%)	24 (24/90)	21 (21/65)	3 (3/22)
>24 h(%)	27 (27/87)	26 (26/65)	1 (1/22)
PIV of Bayesian (mL) (mean + SD) (median) (IQR)	84.734 ± 80.588 (63.924) (6.715, 57.818)	100.382 ± 86.396 (74.882) (25.110, 150.786)	39.291 ± 32.548 (28.562) (19.608, 54.989)
PIV of cSVD(mL) (mean + SD) (median) (IQR)	118.240 ± 109.752 (109.434) (0.335, 8.187)	156.164 ± 101.818 (147.982) (86.763, 209.858)	6.191 ± 11.419 (1.169) (0.476, 6.204)
PIV of oSVD (mL) (mean + SD) (median) (IQR)	65.105 ± 72.614 (49.419) (1.024, 15.897)	84.504 ± 74.319 (60.726) (31.432,119.988)	7.792 ± 12.419 (2.969) (1.883, 8.503)
FIV(mL) (mean + SD) (median) (IQR)	82.930 ± 96.593 (35.320) (9.626, 126.700)	99.870 ± 105.062 (66.770) (10.325, 159.050)	32.879 ± 32.514 (22.11) (8.307, 39.215)

Continuous variables are presented as mean ± SD or median (interquartile range) whereas categorical variables are presented as proportion. NIHSS, National Institutes of Health Stroke Scale; cSVD, cyclic singular value decomposition algorithms; oSVD, oscillatory exponential singular value decomposition algorithms.

**Table 2 diagnostics-13-01810-t002:** The Proportion of Patients with Overestimated Infarct.

	Bayesian	cSVD	oSVD
Progression group	44.6% (29/65)	83.1% (54/65)	49.2% (32/65)
Improvement group	59.1% (13/22)	9.1% (2/22)	9.1% (2/22)
Subgroup 1.a	20.7% (6/29)	6.9% (2/29)	3.4% (1/29)
Subgroup 1.b	58.6% (17/29)	82.8% (24/29)	55.2% (16/29)
Subgroup 2	52.8% (19/36)	83.3 (30/36)	44.4% (16/36)

cSVD, cyclic singular value decomposition algorithms; oSVD, oscillatory exponential singular value decomposition algorithms.

## Data Availability

The data presented in this study are available on request from the corresponding author. The data are not publicly available due to privacy or ethical considerations.

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
