# Peer review of "Comparison of Three Algorithms for Predicting Infarct Volume in Patients with Acute Ischemic Stroke by CT Perfusion Software: Bayesian, CSVD, and OSVD"

_diagnostics, 2023, doi:10.3390/diagnostics13101810_

Round 1

Reviewer 1 Report

This article compares the performance of three algorithms to predict infarct volume in patients with acute ischemic stroke using CT perfusion. The parameters chosen for each of the algorithms are quite different, and I find that to be problematic in making a fair comparison of the performance of the three different algorithms. By default, I would expect the TTP parameter to provide a more accurate result of prediction since TTP calculation involves the contrast signal directly. Tmax on the other hand has to be calculated based on the deconvolution of the arterial input function which will naturally be less accurate due to uncertainities in the voxel location of the AIF and also the venous output. I feel the authors need to improve on their study design to better address this topic. There are also severe language lapses in some sections of the manuscript which made it difficult to understand some parts. Here are some comments that can help for a future submission:
  1. The rCBF, TTP and Tmax thresholds used for comparisons used for comparisons (lines 119-123) are quite different. Especially, looking at oSVD and cSVD, the Tmax and rCBFs are different. Naturally, this would create a discrepance in the predicted volumes. Perhaps those are the default parameters in the software, but to make a fair comparison of the performance of each algorithm, similar values need to be used to assess the predicted volumes. Also, the rTTP>4.5s has been shown to correlate close to  Tmax >6s (PMC5645507). At least, there has to be a reference for choosing these values based on what has been shown in the literature instead of just going with the default values from the system. 
  2. An important issue in this analysis is the designation of the pixels that are used to calculate the arterial input function and venous output especially for oSVD and cSVD since they use Tmax. If the software is selecting this automatically, how can you ensure that the same consistent analysis is done for all three methods across all patients if different pixels get selected for the AIF and VO calculation? A consistent region needs to be selected for a fair analysis, which will affect the accuracy of Tmax. Hence we see that Bayesian had the better performance than the other 2 as TTP does not rely on the AIF or VO.
  3. I have attached some highlighted sections that require language revision. Some of those sections were difficult to understand and grammatically incorrect. 
  4. In Figure 2, the orange and red regions were difficult to decipher. A different color  would be more appropriate to show the difference between the overlap and the core region.
  5. Figure 3 and 5 are low quality.
  6. The  study by reference #22 is very similar to the setup of this study. Why did you not analyze CBF, CBV, MTT maps similarly and the influence of the algorithms on these other maps to better decipher the penumbral region especially in the progressing groups?
  7. Was DWI/PWI MRI performed in some patients to identify the diffusion/perfusion mismatch regions? Just like the later DWI-MRI was used to confirm ischemic core, the initial DWI/PWI mismatch could be used to confirm the extent of penumbra and compare with the algorithms predicted volumes.
  8. Please see other comments in the attached pdf.

Reviewer 2 Report

The authors in the paper compared the performance of the Bayesian probabilistic method, circular Singular Value Decomposition (cSVD), and oscillation index Singular Value Decomposition (oSVD) algorithms in Olea Sphere for predicting infarct volume in patients with acute ischemic stroke (AIS). 90 patients suffering AIS with large vessel occlusion were divided into improvement and progression groups. The improvement group included patients with successful recanalization (TICI 2b-3) after thrombectomy or whose clinical symptoms improved after thrombolysis. The progression group consisted of patients whose clinical symptoms didn't improve. 

The infarct core volume from the Olea Sphere software was used as the predicted infarct volume (PIV) in the improvement group, while the hypoperfusion volume was used as the PIV in the progression group. We defined predicted difference (PD) as PIV minus final infarct volume (FIV) measured at follow-up imaging. Differences among the three algorithms were assessed by the Friedman test. Spearman correlation analysis was used to verify the correlation between PIV and FIV.

This is an interesting comparison of algorithms that evaluate and predict the size of the final stroke core. This would be helpful in qualifying patients, e.g. with tandem lesions, who require anti-platelet therapy during treatment (stent implantation) and selecting patients with the highest risk of a large necrosis area and a high risk of haemorrhage.

Comments:

1/ a very clear downside is the small number of groups, especially the improvement group.

2/ a very large static difference between the groups in clinical characteristics (AF, dyslipidemia, hemorrhagic transformation - is it not a mistake that in the improvement group as many as 11 out of 25 patients had haemorrhagic transformation?),

3/ very large difference between the groups regarding the time from the first symptoms.

I am afraid that this affects the results and does not allow us to accept the obtained results? The results are based on static analysis, so in order not to lose data, propensity score matching should be used to unify the groups.

Reviewer 3 Report

My comments are as follows

1.     Why Bayesian has better predictive power than CSVD and OSVD, Please explain in terms of pathophysiology

2.     Why is there no significant correlation between Bayesian and functional assessment? Does it mean that the area measured by Bayesian is not an area of infarction?

Round 2

Reviewer 1 Report

Changes are acceptable.

Reviewer 3 Report

The author s have replied my comments item-by-item. I have no more comments